# Transcriptional Alterations in X-Linked Dystonia–Parkinsonism Caused by the SVA Retrotransposon

**DOI:** 10.3390/ijms23042231

**Published:** 2022-02-17

**Authors:** Jelena Pozojevic, Shela Marie Algodon, Joseph Neos Cruz, Joanne Trinh, Norbert Brüggemann, Joshua Laß, Karen Grütz, Susen Schaake, Ronnie Tse, Veronica Yumiceba, Nathalie Kruse, Kristin Schulz, Varun K. A. Sreenivasan, Raymond L. Rosales, Roland Dominic G. Jamora, Cid Czarina E. Diesta, Jakob Matschke, Markus Glatzel, Philip Seibler, Kristian Händler, Aleksandar Rakovic, Henriette Kirchner, Malte Spielmann, Frank J. Kaiser, Christine Klein, Ana Westenberger

**Affiliations:** 1Institute of Neurogenetics, University of Lübeck, 23538 Lübeck, Germany; jelena.pozojevic@uksh.de (J.P.); shela.algodon@neuro.uni-luebeck.de (S.M.A.); josephneos1010@gmail.com (J.N.C.); joanne.trinh@neuro.uni-luebeck.de (J.T.); norbert.brueggemann@neuro.uni-luebeck.de (N.B.); joshua.lass@student.uni-luebeck.de (J.L.); karen.gruetz@neuro.uni-luebeck.de (K.G.); susen.schaake@neuro.uni-luebeck.de (S.S.); ronnie.tse@neuro.uni-luebeck.de (R.T.); philip.seibler@neuro.uni-luebeck.de (P.S.); aleksandar.rakovic@neuro.uni-luebeck.de (A.R.); 2Institute of Human Genetics, University of Lübeck, 23538 Lübeck, Germany; veronica.yumicebacorral@student.uni-luebeck.de (V.Y.); nathalie.kruse@uksh.de (N.K.); kristin.schultz@uksh.de (K.S.); varun.sreenivasan@uksh.de (V.K.A.S.); kristian.haendler@uksh.de (K.H.); henriette.kirchner@uksh.de (H.K.); malte.spielmann@uksh.de (M.S.); 3Department of Neurology, University Hospital Schleswig Holstein, 23538 Lübeck, Germany; 4The Hospital Neuroscience Institute, Department of Neurology and Psychiatry and The FMS-Research Center for Health Sciences, University of Santo Tomas, Manila 1008, Philippines; rlrosales@ust.edu.ph; 5Department of Neurosciences, College of Medicine-Philippine General Hospital, University of the Philippines Manila, Manila 1000, Philippines; rgjamora@up.edu.ph; 6Department of Neurosciences, Movement Disorders Clinic, Makati Medical Center, Makati City 1229, Philippines; ciddiesta@gmail.com; 7Institute of Neuropathology, University Medical Center Hamburg-Eppendorf, 20246 Hamburg, Germany; matschke@uke.de (J.M.); m.glatzel@uke.de (M.G.); 8Human Molecular Genomics Group, Max Planck Institute for Molecular Genetics, 14195 Berlin, Germany; 9DZHK (German Centre for Cardiovascular Research), Partner Site Hamburg/Lübeck/Kiel, 23538 Lübeck, Germany; 10Institut für Humangenetik, Universitätsklinikum Essen, Universität Duisburg-Essen, 45147 Essen, Germany; frank.kaiser@uk-essen.de; 11Essener Zentrum für Seltene Erkrankungen, Universitätsmedizin Essen, 45147 Essen, Germany

**Keywords:** XDP, retrotransposon, SVA, splicing, epigenetics, transcription

## Abstract

X-linked dystonia–parkinsonism (XDP) is a severe neurodegenerative disorder that manifests as adult-onset dystonia combined with parkinsonism. A SINE-VNTR-Alu (SVA) retrotransposon inserted in an intron of the *TAF1* gene reduces its expression and alters splicing in XDP patient-derived cells. As a consequence, increased levels of the *TAF1* intron retention transcript *TAF1-32i* can be found in XDP cells as compared to healthy controls. Here, we investigate the sequence of the deep intronic region included in this transcript and show that it is also present in cells from healthy individuals, albeit in lower amounts than in XDP cells, and that it undergoes degradation by nonsense-mediated mRNA decay. Furthermore, we investigate epigenetic marks (e.g., DNA methylation and histone modifications) present in this intronic region and the spanning sequence. Finally, we show that the SVA evinces regulatory potential, as demonstrated by its ability to repress the *TAF1* promoter in vitro. Our results enable a better understanding of the disease mechanisms underlying XDP and transcriptional alterations caused by SVA retrotransposons.

## 1. Introduction

X-linked dystonia–parkinsonism (XDP) is an adult-onset neurodegenerative movement disorder endemic to the Philippines, predominantly affecting men due to the X-linked mode of inheritance. It typically presents in the third to fifth decade of life as a focal dystonia that progresses and becomes generalized, severely incapacitating patients. In patients that survive this disease stage, parkinsonism sets in, overlaps with the dystonia, and predominates from the tenth year of illness onward [1,2]. XDP was initially considered as a pure disorder of the basal ganglia, due to considerable neuronal loss and mosaic gliosis described in the striatum [3,4]. However, more recent findings of reduced cortical thickness and cerebellar gray matter pathology implicate these additional regions in the pathogenesis of XDP [5].

All patients identified to date share a common haplotype, including the likely disease-causing variant, the SVA (SINE-VNTR-Alu) retrotransposon insertion in intron 32 of the *TAF1* gene on the X chromosome [6,7,8,9,10]. Consequently, dysfunction of *TAF1* has been postulated to underlie XDP pathogenesis. Consistent with striatal degeneration, a neuron-specific *TAF1* transcript is reduced in the caudate nucleus of XDP patients, as well as all *TAF1* transcripts in various tissues and cell lines [7,9,10,11,12]. In addition to the reduced *TAF1* expression, the SVA retrotransposon insertion seems to cause increased levels of an alternative splicing isoform, termed *TAF1-32i* and is composed of canonical exon 32 spliced to a cryptic exon within intron 32, terminating 5’ to the SVA insertion [9]. Expression levels of this transcript were found to be higher in XDP cell lines as compared to healthy controls and excision of the SVA restored levels of this transcript in cellular models [9,13]. Furthermore, an inherent part of the SVA is a (*CCCTCT*)_n_ hexamer, where the repeat number n varies among patients (range: 30–55 repeats) and correlates inversely with the age at disease onset, disease severity, and *TAF1* expression [14,15]. Of note, *TAF1* encodes the transcription initiation factor TATA-box binding protein associated factor 1, a subunit of the TFIID complex that mediates transcription by RNA polymerase II, functioning as an important regulator in the expression of a large number of genes [16,17].

Here, we confirm that the alternative *TAF1* splicing isoform, *TAF1-32i*, originally reported only in patient-derived cell lines [9], can be found in cell lines of healthy individuals as well, albeit in significantly lower amounts than in XDP patients. Moreover, we show that this transcript undergoes nonsense-mediated mRNA decay (NMD). By further functional investigations, we observed H3K36me3 within this intronic region, an epigenetic mark present within transcribed regions, while alterations of DNA methylation adjacent to the SVA insertion were not detected. Finally, we found that the SVA alters promoter activity in vitro, suggesting that it may recruit transcription factors or alter chromatin architecture to modulate gene expression.

## 2. Results

### 2.1. The Intron Retention Transcript Is Present in Healthy and XDP Cell Lines

Using patient-derived fibroblasts, induced pluripotent stem cells (iPSCs), and blood samples, we reproduced and confirmed the previous observations that the *TAF1-32i* isoform can be found in various cell lines of patients with XDP (Figure 1a,b, Appendix A) [9,13]. We next investigated the association of *TAF1-32i* expression in the blood of 50 XDP patients with repeat number, age at disease onset (AAO), disease duration, or age at blood collection, and detected no significant correlation (Appendix A). It seems that this alternative transcript variant is also physiologically present in low amounts in non-carriers of the SVA insertion, as we have detected this transcript in various cell lines and tissues, including fibroblasts, iPSCs, and blood samples derived from our healthy controls and SH-SY5Y and HEK293 cells (Figure 1a,b and Appendix A). Our analysis of the levels of *TAF1-32i* in fibroblasts, iPSCs, and blood samples of healthy controls revealed that the expression of this transcript is significantly lower in samples from non-carriers of the SVA insertion when compared to XDP patients (Figure 1a,b). Nevertheless, the ratio between the levels of this transcript in patients and controls was much lower than previously reported [13]. Furthermore, *TAF1-32i* levels were increased in XDP iPSC lines when compared to the iPSC lines in which the SVA was edited out (Figure 1a). Sanger sequencing of the obtained PCR bands revealed that the deep intronic region of this alternative transcript variant, spliced to exon 32, starts at nucleotide position 15,560 from the last nucleotide in exon 32 (Figure 1c). The sequence of the intronic region included in the transcript was identical among different cell types, patients and controls. 

### 2.2. The Intron Retention Transcript Undergoes Nonsense-Mediated mRNA Decay

Given that the *TAF1-32i* transcript is present in low amounts in healthy control iPSCs, we hypothesized that either its synthesis is increased or that its degradation is decreased in XDP. To test whether it undergoes degradation by nonsense-mediated mRNA decay (NMD), we treated the cells with cycloheximide. NMD functions both as an RNA quality control mechanism (via degradation of aberrant transcripts such as those containing disease-causing variants) and a regulator of gene expression (via degradation of normal transcripts, e.g., alternative splicing isoforms), at the interface between transcription and translation (reviewed in [18,19]), and can thus be blocked indirectly by cycloheximide that interferes with protein synthesis. This experiment was performed on healthy control- and patient-derived iPSCs, since the intron retention transcript is the most abundant in this cell type in XDP [9]. We observed increased amounts of this transcript in comparison to untreated cells (2–7-fold changes) upon cycloheximide treatment, both in healthy controls and XDP cells, suggesting that the *TAF1*-*32i* transcript undergoes NMD (Figure 1d). In addition, our results indicate increased synthesis of *TAF1*-*32i* in XDP, given that even upon cycloheximide treatment, levels of this transcript are higher in cells from XDP patients as compared to non-carriers of the SVA insertion.

### 2.3. The Intronic Region Included in the Transcript Is Associated with H3K36me3 in Control and XDP Cells

To further understand the molecular processes underlying the inclusion of the deep intronic region in the transcript, we aimed to investigate epigenetic marks within the region. Visualization of histone modifications from the ENCODE (Encyclopedia Of DNA Elements) project [20] revealed that H3K36me3 (trimethylation of lysine 36 on histone H3) is present in this region, with a particularly strong signal in NT2-D1 and U2OS cells (Figure 2a). Furthermore, chromatin state segmentation, a computational prediction of chromatin states based on chromatin immunoprecipitation-sequencing (ChIP-seq) data, indicated weak transcription in six out of nine cell types. In contrast, this region was annotated as heterochromatic in human embryonic stem cells (H1-hESC), which is generally considered as the cell type most similar to iPSCs (Figure 2a). H3K36me3 is present in gene bodies, marking both exons that undergo active transcription and alternative exons, consistent with its role in alternative splicing [21,22]. Furthermore, it marks constitutive and facultative heterochromatin and plays a role in the DNA damage response by recruiting the DNA repair machinery [23,24]. To experimentally test for the presence of this histone mark in the region, we performed chromatin immunoprecipitation followed by next-generation sequencing (NGS) and qPCR. We did not observe a difference in qPCR results between two healthy controls and three patient-derived iPSC lines, with primers targeting the region predicted to be enriched in this histone mark and included in the transcript (Figure 2b). To cover a wider region of the intron, we performed NGS and confirmed that there was no difference in H3K36me3 levels between a patient and a control line, with a weak signal for this histone mark in intron 32 (Figure 2c).

### 2.4. DNA Methylation Is Not Altered in the 5’ Region Adjacent to the SVA

Given the significance of DNA methylation in alternative splicing regulation and suppression of transposable elements (reviewed in [25,26]), we aimed to investigate CpG methylation of the SVA and the adjacent regions in intron 32. Since the human genome contains >2700 SVA elements [27], we applied long-read nanopore sequencing to precisely target the XDP-specific SVA and bisulfite pyrosequencing to verify the results at selected CpGs in intron 32. We have previously shown that the SVA is heavily methylated in various XDP tissues and cell lines [28]. When comparing the methylation frequency of the regions proximal to the SVA insertion in patients and controls (which also includes the intronic region within the *TAF1*-*32i* transcript), a CpG site at genomic position chrX:70,659,134 (hg19) strikingly deviated from others in brain samples (Figure 3a). Thus, we selected this CpG along with the neighboring one (chrX:70,659,225; hg19) for quantification by pyrosequencing in multiple samples. Our results were consistent while using the two methods, but there was no significant difference in methylation levels between patients and controls across the investigated tissues (Figure 3b). Specifically, although the intron retention transcript is prominent in patient-derived iPSCs, we did not see any difference in methylation levels between healthy controls and XDP patients. The methylation frequency remained high and unchanged even in the “ΔSVA” cell line, an XDP-derived cell line where the SVA was excised by CRISPR/Cas9. These results suggest that increased *TAF1-32i* transcript levels in XDP iPSCs likely cannot be attributed to alterations in DNA methylation.

### 2.5. The SVA Represses TAF1 Promoter Activity In Vitro

An emerging body of evidence indicates that transposable elements function to regulate gene expression by affecting gene transcription, chromatin structure, pre-mRNA processing, and various aspects of mRNA metabolism (reviewed in [29]). SINE retrotransposons can cause epigenetic reprogramming of adjacent gene promoters and can serve as transcriptional enhancers by recruiting various transcription factors [30,31]. Thus, we aimed to test whether the XDP-specific SVA exerts transcriptional activity in a classical enhancer–promoter experiment, using a luciferase assay. First, we characterized the *TAF1* promoter region by inserting either the full-length region (chrX:70,585,177-70,586,242; hg19) or one of its fragments into a promoterless firefly luciferase reporter vector, pGL4.10 (Figure 4a). Promoter fragments were created based on histone marks and DNase sensitivity in order to define the most active region. In this experimental setup, the *TAF1* promoter controls expression of the luciferase gene, and any change in the promoter activity will be detected as a change in the luciferase signal. Measurement of relative luciferase activity upon transfecting HEK293 cells with these constructs narrowed down the most active *TAF1* promoter region to a fragment of approximately 400 bp (chrX:70,585,696-70,586,107; hg19) (Figure 4b). Next, we inserted the full-length SVA containing the hexanucleotide repeat with the minimum reported number of units ((*CCCTCT*)_30_) in either sense or antisense orientation into the pGL4.10 vector containing this 400 bp *TAF1* promoter region. Our results show that the SVA (inserted in either direction) strongly suppresses *TAF1* promoter activity in comparison to a size-matched control, suggesting its regulatory potential and possible recruitment of transcription factors (Figure 4c).

## 3. Discussion

TAF1 is the largest subunit of TFIID, the initial basal transcription factor that recognizes and binds to the core promoter, and is thus essential for the subsequent formation of the functional preinitiation complex that positions RNA polymerase II at transcription start sites (reviewed in [32]). Furthermore, *TAF1* is extremely intolerant to loss-of-function (LoF) mutations (probability of LoF Intolerance, pLI = 1 in gnomAD database), and a *taf1* knockout zebrafish model shows embryonic lethality, pointing to its crucial role in development [33]. Together, this implies that the expression levels and function of *TAF1*/TAF1 must be tightly regulated, and any dysregulation could have a plethora of different consequences. In addition, this is a large gene with numerous transcript variants (currently, 27 annotated in Ensembl) that contributes to both proteomic diversity and to the tissue-specific gene regulatory network. While the canonical and neuron-specific *TAF1* isoforms differ in only 6 bp that determine the tissue distribution (e.g., neuronal commitment), it is still unclear whether the alternative transcript variant containing the deep intronic region performs a specific role in the cell. Given its increased amounts in XDP cells, it is tempting to speculate that it might exert a dominant-negative effect if it is being translated into a protein or RNA-induced toxicity and accumulation/deposition of RNA-binding proteins.

Our results show that although the level of the *TAF1-32i* transcript differs significantly between XDP patients and controls, the retained region within intron 32 is identical in all individuals. However, we do not know where the transcript terminates, or whether it even includes a part of the SVA in XDP patients. SVA retrotransposons have been reported to cause aberrant splicing, altering the canonical transcripts [34,35,36]. Instead, the XDP-specific SVA seems to enhance the synthesis of an already existing transcript, as we demonstrated that low levels of the *TAF1*-*32i* transcript can be detected in various cell types and that it undergoes degradation by NMD. Recent evidence indicates that coordinated action between alternative splicing and NMD functions to achieve the proper expression level of a given gene and/or protein, and that intron retention may be used to regulate a specific differentiation event, as shown in the hematopoietic system [37,38]. Thus, it seems plausible that levels of the *TAF1*-*32i* transcript might fine-tune expression, cellular differentiation, or cellular decisions, and that altered levels could contribute to disease manifestation. Of note, our experiments suggest that the levels of *TAF1*-*32i* in non-XDP cells might be higher than previously estimated when analyzed directly from cDNA (i.e., without prior preamplification).

In accordance with its role in marking transcribed regions, we detected a weak H3K36me3 signal in the deep intronic region included in the *TAF1-32i* transcript. This histone mark is also associated with the binding of PTBP (polypyrimidine tract-binding-protein), one of the major regulators of splicing that was shown to bind to silencing elements and regulate whether or not an alternative exon will be included in a transcript [21]. Although there were no differences in H3K36me3 levels between patients and controls along the transcribed regions, we cannot conclude that there is no signal within the SVA. Namely, with short-read sequencing technologies, it is challenging to map a putative H3K36me3 signal coming from any SVA, even if it is being transcribed (at least partially). That is, because there are >2700 SVA elements in the human genome, they are being filtered out during the short-read bioinformatical analysis that includes only the regions that can be mapped to the reference sequence. Conversely, for estimating DNA methylation, long-read sequencing technologies exist and enable the measuring of DNA methylation along the region spanning the SVA insertion of interest. Therefore, we chose to use the nanopore technology that detects native DNA modifications. Subsequently, we applied single-nucleotide-specific pyrosequencing, which is methodologically different and relies on bisulfite conversion prior to measurement. Although differences in *TAF1*-*32i* amounts are prominent between controls and XDP lines in iPSC (Figure 1), these differences are not caused by changes in DNA methylation, as demonstrated by our results (Figure 3a,b). Specifically, these differences in levels of the *TAF1-32i* transcript are visible in XDP ΔSVA cells (Figure 1a), which do not coincide with any alterations in DNA methylation at these two CpG sites (Figure 3b). DNA methylation levels at the position chrX:70,659,134 appear to vary drastically among tissues, with the lowest levels in the cerebellum, potentially indicating a tissue-specific regulatory effect. Although we have not observed any alterations in DNA methylation and H3K36me3 levels in the intronic region within *TAF1-32i* in XDP-derived samples, further investigations of other histone marks and other CpG sites are warranted. For instance, recent work on XDP-derived cells reported local changes in H3 acetylation (AcH3), affecting an exon proximal to the SVA insertion. This decrease in AcH3 level was normalized by CRISPR/Cas9-excision of the SVA, suggesting that the SVA alters epigenetic marks in the region [39]. In addition, a significant increase in histone H3 citrullination (H3R2R8R17cit3) was reported in the XDP post-mortem prefrontal cortex [40]. When considering XDP-relevant epigenetic changes beyond those potentially introduced by the SVA, the three disease-specific single-nucleotide changes introduce or abolish CpG sites of DNA methylation, introducing a possible additional mechanism that might modulate *TAF1* expression in XDP in addition to the SVA [41].

Transposable elements comprise a large portion of the human genome, and in healthy cells, they are silenced and usually inactive. However, they have been reported to become active and mobile in aging mammalian tissues [42] and are regarded as a source of genomic variation or even as “controlling elements” [43]. Currently, it is well known that transposable elements can influence gene expression by acting as promoters, enhancers, repressors, or insulators (reviewed in [44]). Our results showing that the XDP-specific SVA retrotransposon represses *TAF1* promoter activity (Figure 4c) suggest that it could function as a transcriptional repressor. This is in line with a previous report investigating an SVA element inserted upstream of the *FUS* gene and is thus associated with amyotrophic lateral sclerosis and frontotemporal dementia. Namely, this SVA exerts a repressive function on the SV40 minimal promoter [45], indicating that this property is universal rather than sequence-dependent. Conversely, the XDP-specific SVA was shown to act as a promoter in a different experimental setup [15], also leading to the conclusion that there are transcription factor binding sites (TFBS) within the SVA. However, due to its repetitive sequence and genome-wide distribution, it is still experimentally challenging to prove the exact transcription factor(s) and their binding sites within the SVA. Although our study was limited in some aspects, such as small sample size, that did not always allow for statistical testing, it adds to the growing body of evidence that transposable elements affect gene expression. Together, our results show transcriptional alterations in XDP caused by the SVA retrotransposon, suggesting that it contains binding sites for transcription factors and possibly splicing regulators.

## 4. Materials and Methods

### 4.1. Study Participants

We analyzed biomaterials from a total of 83 individuals (52 XDP patients carrying the SVA insertion and 31 (21 males) healthy ethnicity-matched controls with wild-type genotype). The median age at sample collection was 40 (interquartile range (IQR: 35.0–47.8, range: 30–60) years for the XDP patients and 35 (IQR: 30.0–42.0, range: 18–54) years for controls. In XDP patients, the median repeat number was 43 (IQR: 40.0–45.8, range: 35–53), median AAO was 37 (IQR: 31.0–41.8, range: 26–51) years, and median disease duration at the time of sample collection was 3 (IQR: 2–4, range: −2–19) years.

DNA was available for all samples. With respect to RNA, for 41 XDP patients, only RNA from blood was available and used in experiments, while for two patients, only RNA from fibroblasts was available. For the remaining patients, RNA from the blood and/or fibroblasts or iPSCs was available. For two XDP patients and 3 controls, postmortem brain tissue was available. The XDP patients died at the age of 36 and 38 years. Two of the controls were male and one female, and they were 68 years old at the time of death, with no neurodegenerative findings at the time of pathological examination. They were of German ethnicity, as no postmortem tissue from Filipino individuals was available. The study was conducted according to the guidelines of the Declaration of Helsinki and approved by the Ethics Committee of the University of Lübeck (AZ12-219). All autopsies had been performed either as clinical autopsies with first-line relatives, next of kin or their legally authorized representatives giving informed consent or as legal autopsies on behalf of investigating authorities. The use of specimens obtained at autopsies for research upon anonymization is in accordance with local ethical standards and regulations at the University Hospital Schleswig–Holsten (the “Gesetz uber das Leichen–, Bestattungs– und Friedhofswesen (Bestattungsgesetz) des Landes Schleswig–Holstein vom 04.02.2005, Abschnitt II, 9 (Leichen offnung, anatomisch)”) or University Medical Center Hamburg–Eppendorf (“Hamburgisches Krankenhausgesetz vom 17.04.1991, §12, Abs. 1”).

### 4.2. Nucleic Acid Extraction and Reverse Transcription

The genomic DNA was routinely extracted from peripheral blood leukocytes using the salting-out method. QIAamp DNA Mini Kit (Qiagen, Hilden, Germany) was used for DNA extraction from iPSCs, while the Blood and cell culture DNA midi kit (Qiagen) was used to extract high-molecular-weight DNA from brain tissue (i.e., cerebellum and frontal cortex). RNA was extracted from the whole blood using the PAXgene Blood RNA Kit (Qiagen), and from cells using the RNeasy Mini Kit (Qiagen), according to the manufacturer’s instructions. Only the RNA samples with an RNA integrity number (RIN) of >6 were included in the analyses. Maxima First Strand cDNA Synthesis Kit for RT-qPCR with dsDNase (Thermo Scientific, Waltham, MA, USA) was used for reverse transcription, starting with 500 ng total RNA. 

### 4.3. Quantitative PCR (qPCR)

Maxima SYBR Green/Fluorescein qPCR Master Mix (Thermo Scientific) was used for qPCR, in a 10 μL reaction volume, on the Light Cycler 96 Instrument (Roche, Basel, Switzerland). *TAF1*-*32i* primers target exon 32 (5′-GTATAATGATTCAGGAAGTTGCAAG-3′) and intron 32 (5′-GTAATGTACCAATATAAATTTCCTGGTTT-3′). *GAPDH* primers target exon 1 (5′-GTCAGCCGCATCTTCTTTTG-3′) and exon 3 (5′-GCGCCCAATACGACCAAATC-3′). Cycling conditions for the 3-step amplification are as follows: 95° for 15 s; 57° for 30 s; 72° for 30 s. The analysis of each sample was performed in triplicate. Statistical analyses were performed on dCt values (Ct target–Ct reference gene) using an unpaired *t* test since they are logarithmic and thus normally distributed, which allows for one to perform a *t* test.

### 4.4. Sanger Sequencing

PCR products using cDNA as a template and the above-given sequences of the *TAF1* in32 primers were purified by Exonuclease I and Fast AP Thermosensitive Alkaline Phosphatase (Thermo Scientific). Subsequently, the sequencing reaction was performed using only one of the primers and the BigDye Terminator v3.1 Cycle Sequencing Kit (Applied Biosystems, Waltham, MA, USA). Samples were purified by Sodium Acetate/Ethanol precipitation, dissolved in Hi-Di Formamide (Applied Biosystems), and loaded on the 3500xL Genetic Analyzer (Applied Biosystems). Electropherograms were visualized using Chromas Lite (Technelysium Pty Ltd., South Brisbane, Australia).

### 4.5. Cell Culture

Fibroblast lines were established from skin biopsies, and these cells, as well as HEK293, were grown in DMEM medium (Thermo Scientific, Waltham, MA, USA), supplemented with 10% fetal bovine serum (Thermo Scientific, Waltham, MA, USA) and 1% Penicillin–Streptomycin (Thermo Scientific, Waltham, MA, USA). Generation and characterization of the iPS cell lines from XDP patients and ethnically matched controls was performed previously (https://www.wicell.org/home/stem-cells/catalog-of-stem-cell-lines/collections/massachusetts-general-hospital.cmsx (accessed on 10 December 2021)), and gene-edited lines have been examined in an earlier study [10]. Here, they were grown on Matrigel-coated plates in mTeSR medium (StemCell Technologies, Vancouver, BC, Canada).

### 4.6. Cycloheximide Treatment

iPSCs from healthy controls and XDP patients were grown to 70% confluency in 6-well plates, prior to cycloheximide treatment (C4859, Sigma-Aldrich, St. Louis, MO, USA). On the day of the treatment, the medium was removed from the cells, and fresh medium containing cycloheximide to a final concentration of 50 μg/mL was added to each well. The cells were incubated overnight and pelleted the next day for further experiments (i.e., RNA extraction and qPCR).

### 4.7. Chromatin Immunoprecipitation (ChIP)

Chromatin immunoprecipitation was performed according to Lee et al. [46]. Briefly, iPSCs were fixed for 10 min on ice with 1% formaldehyde in mTeSR. The reaction was quenched with 2.5 M glycine, followed by extraction of the nuclear lysate and chromatin sonication on Diagenode Bioruptor Pico (15 cycles; 30 s pulse–30 s pause). For ChIP, 15 μg of chromatin was incubated with 5 μg of the H3K36me3 antibody (ab9050, Abcam, Cambridge, UK) overnight. The next day, blocked magnetic beads were added to the chromatin–antibody complexes and incubated overnight, followed by 7 washes with RIPA buffer and 1 with TE buffer. The immunoprecipitated DNA was extracted with phenol–chloroform, washed with ethanol, eluted in ultra-pure nuclease-free water, and used for subsequent experiments. For qPCR, the primers were designed to cover both the enriched H3K36me3 signal (Figure 2a) and the region included in the *TAF1*-*32i* transcript (forward: 5′-GCTCATGAATGTATTCTGATCC-3′; reverse: 5’-GTACAGCTATGTAAGATATTGCC-3′). For NGS, library preparation was performed with NEBNext Ultra II DNA Library Prep with Sample Purification Beads (E7103, NEB), and the sequencing was performed on NextSeq 2000 (Illumina, San Diego, CA, USA). Reads were mapped using Bowtie2, while parsing to bigwig format was performed using DROMPAplus and/or MACS2 (as described in [47]).

### 4.8. DNA Methylation Analyses by Nanopore Sequencing

Cas9-targeted sequencing from Oxford Nanopore Technologies was performed to enrich the target region and to obtain the epigenetic information. CRISPR RNAs (crRNAs) were designed with CHOPCHOP (https://chopchop.cbu.uib.no (accessed on 10 December 2021)). Four crRNAs were used upstream of the *TAF1* SVA insertion, and four crRNAs were used downstream. Two libraries were prepared per sample. The enriched DNA was prepared with the Nanopore Ligation Sequencing Kit (SQK-LSK109), loaded on a R9.4.1 flow cell and sequenced with MinION or GridION. For methylation analysis, all sequencing data obtained were combined to maximize coverage depth. Methylation was called with the software Nanopolish (v0.13.2) (Oxford Nanopore Technologies, Oxford, UK), which can detect 5’-methylcytosine (5mC) in a CpG context. To counteract potential off-target effects of the CRISPR/Cas9 enrichment, the BAM file was filtered for reads with an alignment length >3kb in the patient- or >1.5kb in control-derived samples. Only CpG sites covered by >10 reads were included in the analysis.

### 4.9. DNA Methylation Analyses by Pyrosequencing

Bisulfite conversion of genomic DNA was performed with the EpiTect Fast DNA Bisulfite Kit (Qiagen), according to the manufacturer’s instructions. Converted DNA was PCR amplified using primers specific for the converted sequence (forward: 5′-ATAATTTTTAATTTGGGTTTAATGGGG-3′; reverse: 5′-[BIO]CTACCTAACAAAAATATAAATAATAAATTAA-3′). Samples were sequenced on PyroMark Q48 Autoprep (Qiagen) using two different sequencing primers (70659134: 5’-GTATTAATATTATTTAGTAGTT-3’; 70659225: 5’-GTTTATATTATATTTTGTTTAG-3’). Data were tested for normal distribution using the Kolmogorov–Smirnov test and analyzed with an unpaired *t* test.

### 4.10. Luciferase Assay

To define the most active *TAF1* promoter region, various fragments were inserted into the pGL4.10(*luc2*) vector (Promega, Madison, WI, USA) using the Gibson Assembly cloning strategy (E2621, NEB). In the next step, either the full-length SVA or a size-matched control were inserted in the vector containing the most active *TAF1* promoter region. Primers used for cloning are available upon request. To improve the cloning efficiency of the repetitive DNA regions found within the SVA (e.g., hexanucleotide repeats), OneShot Stbl3 chemically competent *E. coli* were used (Invitrogen, Waltham, MA, USA), and bacteria were grown at 30 °C. In addition to Sanger sequencing, inserts were verified with fragment analysis targeting the hexanucleotide repeats, as described previously [14,15]. HEK293 cells were transfected with different constructs, using FuGENE HD (Promega). We co-transfected cells with a thymidine kinase promoter-Renilla luciferase reporter plasmid (pRL-TK) as an internal control. After 24 h, the cells were lysed, and the activity of Firefly and Renilla luciferase was determined with the Dual Luciferase Reporter Assay (Promega) in a TriStar2 LB Multidetection Microplate Reader (Berthold, Bad Wildbad, Germany). All measurements were verified in at least three independent experiments and as triplicates in each experiment. Firefly luciferase signals were corrected for transfection efficiency using Renilla signals of the co-transfected control vector. Relative light units were then normalized relative to the *TAF1* pro 400 bp-containing plasmid. Statistical analysis was performed using the Kruskal–Wallis test with Dunn’s multiple comparison test.

## Figures and Tables

**Figure 1 ijms-23-02231-f001:**
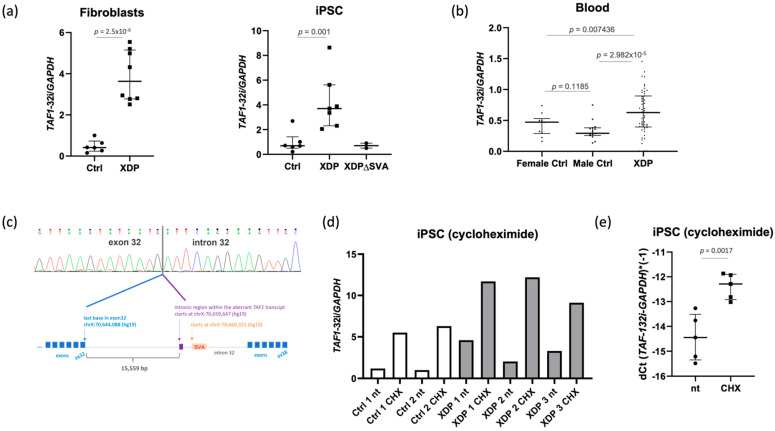
*TAF1*-*32i* transcript in cells of healthy controls and patients with XDP. (**a**) qPCR results of fibroblasts (*n* = 6 controls; *n* = 8 patients) and induced pluripotent stem cells (iPSC; *n* = 6 controls; *n* = 7 patients; *n* = 2 XDPΔSVA cell lines), normalized to *GAPDH* levels. XDPΔSVA refers to the patient-derived cell lines where the SVA was excised by CRISPR/Cas9. *t* test was performed on dCt values. (**b**) qPCR results (*n* = 10 female controls; *n* = 15 male controls; *n* = 50 patients) on blood-derived cDNA samples, relative to *GAPDH*. *p* values result from pair-wise Wilcoxon rank-sum test. To overcome possible batch effects, two independent samples were measured repeatedly in each batch, and all other samples were corrected according to the mean changes in measurement for these two samples. (**c**) Sanger sequencing showing the sequence of the intronic region included in the transcript and the scheme explaining the genomic locations, with genomic coordinates in the hg19/GRCh37 assembly. Blue squares represent canonical *TAF1* exons, the violet square represents the intronic region within the transcript (not drawn to scale). (**d**) qPCR results showing increased levels of the *TAF1-32i* transcript in control (*n* = 2) and patient-derived cells (*n* = 3) after cycloheximide (CHX) treatment, as compared to the non-treated (nt) cells. (**e**) *t* test on dCt values, comparing non-treated and CHX-treated samples. Ctrl, control.

**Figure 2 ijms-23-02231-f002:**
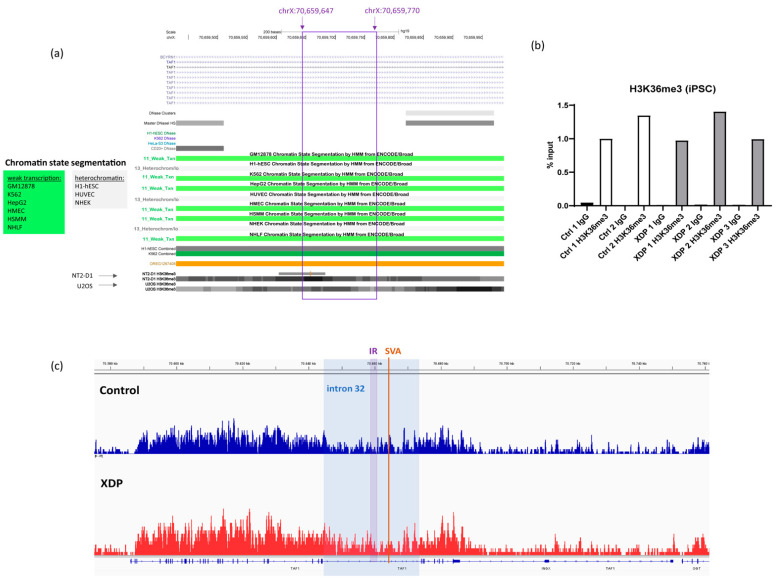
H3K36me3 levels at the intronic region included in the transcript. (**a**) USCS browser screenshot, showing chromatin state segmentation and H3K36me3 histone mark in different cell lines. Note that the strongest signal is present in NT2-D1 cells. The region marked in violet is included in the transcript, and its genomic coordinates are shown above. (**b**) ChIP-qPCR results from control (*n* = 2) and XDP (*n* = 3) iPSC lines. Results are calculated relative to the corresponding input sample and shown as % input. (**c**) ChIP-seq results showing the *TAF1* locus in control (blue) and XDP-derived (red) iPSCs. The region highlighted in blue depicts *TAF1* intron 32; the narrower region marked as IR indicates the intronic region retained in the *TAF1-32i* transcript, and the orange line marks the position in which the SVA is inserted.

**Figure 3 ijms-23-02231-f003:**
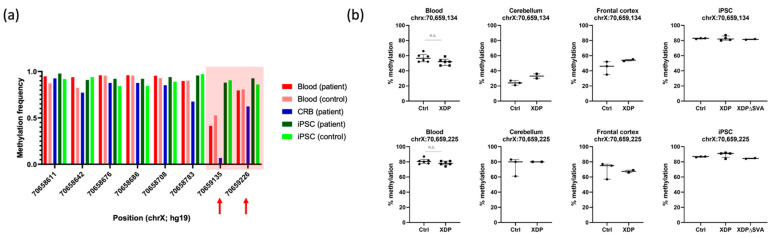
DNA methylation in intron 32 of *TAF1*, proximal to the SVA insertion. (**a**) Long-read nanopore sequencing on DNA from different tissues and cell lines from a healthy control and a patient with XDP (i.e., blood, iPSCs, cerebellum). Arrows indicate the two CpGs selected for analysis by pyrosequencing. (**b**) Pyrosequencing results showing methylation levels at the two selected CpGs: chrX:70,659,134 and chrX:70,659,225 (hg19), in blood (*n* = 6 controls, *n* = 6 patients), cerebellum (*n* = 3 controls; *n* = 2 patients), frontal cortex (*n* = 3 controls; *n* = 2 patients), and iPSCs (*n* = 3 controls; *n* = 4 patients; *n* = 2 XDPΔSVA cell lines). Unpaired *t* test was performed on blood samples after testing for normality with Kolmogorov–Smirnov test. n.s., not significant.

**Figure 4 ijms-23-02231-f004:**
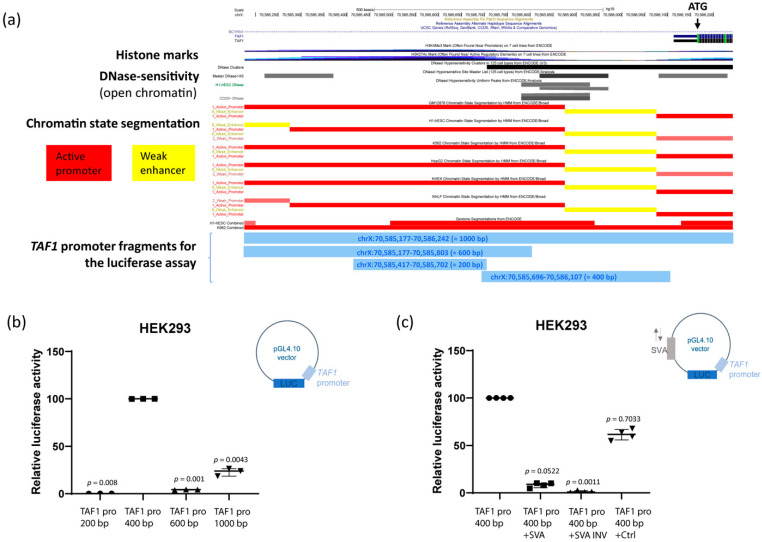
Regulatory potential of the SVA in vitro. (**a**) UCSC browser screenshot of the *TAF1* promoter region, showing DNase sensitivity, histone modifications and chromatin state segmentation in various cell types. In light blue (the lower part of the figure) are shown promoter fragments that were investigated in luciferase reporter assays, along with their genomic coordinates (hg19). Note that the most active *TAF1* promoter region (fragment ≈ 400 bp) overlaps with the DNase-sensitive region, characteristic of open chromatin. (**b**) Relative luciferase activity (Firefly counts/Renilla TK counts) of different *TAF1* promoter regions define the most active region (Fragment ≈ 400 bp). (**c**) Relative luciferase activity of the XDP-specific SVA-inserted sense or antisense (INV-inverted), as compared to a size-matched control. Maximum activity (100%) was exerted by the vector containing only the most active promoter region (*TAF1* pro 400 bp), and *p* values are calculated relative to this sample using Kruskal–Wallis and Dunn’s multiple comparison tests.

## Data Availability

The data presented in this study are available from the corresponding authors upon reasonable request.

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
