# Peer review of "Transcriptional Alterations in X-Linked Dystonia–Parkinsonism Caused by the SVA Retrotransposon"

_ijms, 2022, doi:10.3390/ijms23042231_

Round 1
Reviewer 1 Report
Pozojevic and collaborators describe the molecular characterization of the involvement of the SVA retrotransposon in the XDP gene. This is a thorough analysis of the transcriptional regulation of the XDP gene and the effect of the SVA and of the hexanucleotide expansion in the SVA on it.
Although the authors perform a complete analysis of several molecular mechanisms affecting/affected by the presence of the SVA retrotransposon, some missing information could be relevant to explain, or contribute to explain, part of the results.
The most relevant point is the characterization of the expansion in this element. This hexanucleotide expansion within SVA has been shown, as correctly referenced by the authors, as responsible for the presence of the disease but also with some clinical characteristics such as age at onset. Under these circumstances it is necessary to characterize, or to report if its already known, the size of the expansion and if the different lines have different sizes, include that variable in the analysis.
Minor points.
- In figure 1, authors should add a brief note stating that the gene/exons/introns are not drawn to scale.
- In figure 2C it would be helpful to have an indication of whether the SVA transposon lies.
Reviewer 2 Report
This article does not have sufficient scientific advances.
The authors should approve the experiments and present more significant results.
Author Response
We would like to thank the reviewer for taking the time to review our manuscript. There were no specific questions/comments to address.Reviewer 3 Report
Please find attached the general and specific comments and suggestions.

Author Response
Please find attached our response

Reviewer 4 Report
This study is focused on the alternative TAF1 splicing isoform (TAF1-32i), the SVA retrotransposon and their implication in X-linked dystonia-parkinsonism (XDP). Here, it was found the TAF1-32i transcript was higher in XDP cells compared to healthy cells, and when the SVA was removed the levels of TAF1-32i were restored. Then the authors found an increase of this transcript after cycloheximide treatment both in healthy and XDP patient-derived iPSCs, suggesting that this transcript undergoes nonsense-mediated mRNA decay (NMD). Then, it was demonstrated that the increase on the TAF1-32i transcript in XDP iPSCa is not due to alteration in DNA methylation. Finally, it was demonstrated that SVA alters the activity of promoter in vitro. The methodology used for acquiring data is appropriate and authors have provided sufficient literature to cover the critical background information. The paper presents interesting results; however, the authors should address next points:
- Is there significant difference between Control vs XDP groups both in fibroblast and iPSC?
- In section 2.2, add the percentage increase of the transcript on treated cells (results description).
- In section 4 add the statistics methods used as well as in each figure legend.
Round 2
Reviewer 1 Report
The authors have adequately addressed my previous concerns. I have no additional comments at this time.
Reviewer 2 Report
I thank the authors for their great efforts to improve the revised article by following the reviewer's comments.
Now, I have reevaluated the manuscript and recommend the publication of IJMS.